# The Japanese encephalitis virus NS1' protein facilitates virus infection in mosquitoes

Yuwei Liu[1], Yutian Huang[1], Ruidong Li[1], Chang Miao[1], Yi He[1], Changhao Xu[1], Xi Zhu[1], Bowen Li[1], Rui Wu[1,2,3], Qin Zhao[1,2,3], Yiping Wen[1,2,3], Xiaobo Huang[1,2,3], Qi-gui Yan[1,2,3], Yi-fei Lang[1,2,3], Shan Zhao[1,2,3], Yiping Wang[1,2,3], Yajie Hu[4], San-jie Cao[1,2,3], Senyan Du[1,2,3]*

**1** Research Center for Swine Diseases, College of Veterinary Medicine, Sichuan Agricultural University, Chengdu, China, **2** Sichuan Science-Observation Experimental Station for Veterinary Drugs and Veterinary Diagnostic Technology, Ministry of Agriculture, Chengdu, China, **3** Engineering Research Center of Southwest Animal Disease Prevention and Control Technology, Ministry of Education of the People's Republic of China, Chengdu, China, **4** Sichuan Center for Disease Control and Prevention, Chengdu, China

☯ These authors contributed equally to this work.
* senyandu@sicau.edu.cn

## Abstract

### Background

The Japanese encephalitis virus (JEV), a mosquito-borne flavivirus, is known for its capacity to cause severe neurological disease in Asia. Neurotropic flaviviruses within the Japanese encephalitis (JE) serogroup possess the distinctive feature of expressing a unique nonstructural protein, NS1´. The NS1´ protein consists of the full NS1 protein with an additional 52 amino acid extension at the C-terminus and has been demonstrated to exhibit virulence in mammalian hosts upon infection. However, the precise role of the NS1´ protein in the mosquito vectors has yet to be elucidated.

### Methodology/principal findings

In this study, an NS1´-defective virus (rG66A) was engineered, and its effect on the infection of mosquito cells was investigated. The results demonstrated a significant reduction in the infectivity of the rG66A virus in mosquito cells by RT-qPCR, indicating that the absence of the NS1´ protein impedes JEV replication in *Culex* mosquitoes. Additionally, this research elucidated the underlying mechanism by which the NS1´ protein enhances viral infection in mosquitoes by RNA-Seq analysis. Specifically, the NS1´ protein was found to facilitate infection through the suppression of antimicrobial peptides (AMPs) regulated by the Toll pathway.

### Conclusions/significance

Our research demonstrated that the JEV NS1´ protein contributes to immune escape, thereby enhancing viral infection in mosquitoes. This finding offers new insights into the transmission mechanisms of JEV, elucidating novel aspects of viral propagation.

**Data availability statement:** All relevant data are within the manuscript and its Supporting Information files. The sequencing data associated with this project have been deposited in the NCBI Short Read Archive database (accession number: PRJNA1193468).

**Funding:** This work was supported by the Natural Science Foundation of Sichuan Province (2024NSFSC1269 to SD), National Natural Science Foundation of China (32102748 to SD) and National College Student Innovation and Entrepreneurship Training Program (No. 202310626032 to SD). The funders had no role in study design, data collection and analysis, decision to publish, or preparation of the manuscript.

**Competing interests:** The authors have declared that no competing interests exist.

## Author summary

Japanese encephalitis virus (JEV) is the primary etiological agent of acute human encephalitis in Asia, and currently, there is no specific antiviral therapy for JE. JEV is transmitted to mammalian via the bite of an infected mosquito, making the acquisition of the virus from an infected mammalian host by the vector a critical step in the transmission cycle. JEV produces a distinctive nonstructural protein, NS1′, which is produced through a −1 ribosomal frameshift. Previous studies have demonstrated that the NS1′ protein plays a crucial role in the virulence of both West Nile virus (WNV) and JEV. Our research demonstrates that the NS1′ protein facilitates JEV infection in mosquitoes by suppressing their innate immunity response. This discovery elucidates a previously unrecognized mechanism of JEV infection, offering potential avenues for the development of innovative prevention strategies against JEV.

## Introduction

Japanese encephalitis virus (JEV) belongs to the *Flavivirus* genus which is recognized as a significant group of mosquito-borne viruses. JEV predominantly circulates in the southeastern and southern regions of Asia [1,2], with occasional cases reported in northern Australia and parts of the Western Pacific [3–6]. Nevertheless, the effects of global warming and urbanization have facilitated the spread of JEV to additional locations such as Australia, Pakistan, and Saipan [7–9]. Notably, JEV infects individuals across all age groups, with children serving as the most affected hosts, while adults who were exposed to the pathogen during childhood possess natural immunity. The occurrence of JEV outbreaks not only presents a potential threat to the well-being of both animals and humans, but also exerts a substantial influence on the agricultural production economy. To advance our understanding of JEV control measures, it is imperative to increase our understanding of the disease burden at the national level. The transmission cycle of JEV primarily involves mammals and mosquitoes, particularly those belonging to the *Culex* species. Although the role of mosquitoes in the transmission and infection of JEV is of paramount significance, our current knowledge regarding the mechanism through which JEV infects mosquito vectors remains incomplete. Therefore, it is crucial to determine a thorough comprehension of the interplay between JEV and mosquito vectors in order to effectively control the transmission of JEV.

The genome of JEV consists of a single-stranded positive-sense RNA approximately 11kb in length that encodes a unique polyprotein. This polyprotein undergoes cleavage by both host and viral proteases, resulting in the production of three structural proteins (C, prM, and E) and seven nonstructural proteins (NS1, NS2A, NS2B, NS3, NS4A, NS4B, and NS5) [2,3]. Among these proteins, NS1 serves as a multifunctional glycoprotein that plays a role in viral replication [10–12], modulation of the immune response [13–15], and facilitation of viral acquisition by mosquito vectors [16]. An additional nonstructural protein, NS1′, which is a 52 amino acid C-terminal extension of NS1, is exclusively produced by members of the JE serogroup as a result of −1 programmed ribosomal frameshift (PRF) [17,18]. Further investigation revealed that NS1′ shares a similar cellular distribution with NS1 [19]. Moreover, experimental evidence suggests that the NS1′ protein plays a significant role in the virulence of both WNV and JEV [20–22]. The elimination of NS1′ has been shown to attenuate neurovirulence in both WNV and JEV. In WNV, a single amino acid substitution in NS2A (A30P) dramatically resulted in the complete cessation of NS1' protein production and an increase in the production of IFN-α/β [18,20,21,23,24]. Previous studies have indicated that the JEV NS1′

protein inhibits the production of IFN-α/β by targeting MAVS and facilitates viral replication within host cells [25]. Additionally, JEV NS1′ has been found to interact with the host CDK1 protein and regulate the antiviral response [26]. Moreover, the JEV NS1′ protein promotes persistent viral infection in pig tonsils by targeting dendritic cells and macrophages [27]. Furthermore, the JEV live attenuated vaccine strain SA14-14-2, which lacks NS1′, exhibits heightened virulence upon reintroduction of NS1′ [22]. Other evidence has also demonstrated that the presence of NS1′ in avian cells enhances JEV production [28]. The presence of NS1′ can be detected in the mosquito cell lines [27]. While the roles of NS1′ have been previously demonstrated in mammalian cells, the function of JEV NS1′ in mosquito vectors remains unexplored and the mechanism by which it contributes to virus transmission remains unclear.

Unlike the immune system of mammals, the immune system of mosquitoes relies on innate immune mechanisms to combat viral infections. Numerous studies have demonstrated that RNA interference (RNAi) and other evolutionarily conserved innate immune responses, such as the Janus kinase-signal transducer and activator of transcription (JAK-STAT), Toll, and immune deficiency (Imd) pathways [29–33], play crucial roles in systemic defense against viral infections. The activation of these innate immunity pathways upon viral infection leads to the transcription of antiviral genes. Additionally, the Toll and Imd pathways induce the upregulation of antimicrobial peptides (AMPs), which exhibit potent antiviral effects. Insects possess four distinct families of AMPs, which can be classified based on their unique sequences and structural characteristics [34,35]. It is frequently observed that defensins, cecropins, proline-rich peptides, and attacins are commonly present in mosquitoes. The majority of AMPs are known to possess antibacterial and/or antifungal properties. However, recent research indicates that certain AMPs also exhibit antiviral activity [36–38].

In this study, we generated a defective NS1' virus (rG66A) to elucidate the role and mechanism of the NS1' protein in mosquito vectors. Our findings provide evidence that the NS1' protein enhances JEV infection in both mosquito cell lines and vectors. Mechanistically, we demonstrated that the NS1' protein suppresses the Toll, and downregulates the expression of AMPs. These findings underscore the critical function of the NS1' protein in JEV infection in mosquito vectors.

## Materials and methods

### Ethics statement

Mice blood samples were collected with the approval of the local ethics committee at Sichuan Agricultural University. All animal experiments were carried out in accordance with the protocols approved by the Institutional Animal Care and Use Committee of Sichuan Agricultural University (Approval No. 20210034).

### Mosquitoes, cells and viruses

*Culex quinquefasciatus* (the Chengdu strain) was reared at 28 °C and 80% humidity. The BHK21 cells and C6/36 cells were maintained in Dulbecco's Modified Eagle's Medium (DMEM, Gibco) supplemented with 10% FBS (ExCell Bio) and 1% antibiotic-antimycotic solution (Thermo). The *Drosophila melanogaster* S2 cell line was cultured in Schneider's medium (Gibco) supplemented with 10% heat-inactivated fetal bovine serum and 1% antibiotic-antimycotic solution. Cxq cells were derived from newly hatched *Culex quinquefasciatus* larvae and were a gift from Dr. Feng Cui [39]. The BHK21 and C6/36 cell lines were purchased from ATCC. The *Drosophila* S2 cell line was provided by Invitrogen. JEV (SA14 strain, MK585066.1) was maintained in our laboratory and propagated in C6/36 cells. The

NS1′ defective viruses (rG66A, a "G'-to-'A' mutation at nucleotide position 66 in the NS2A gene of SA14 of JEV) were produced by electroporation of BHK21 cells with transcribed RNA from the full-length cDNA clones pACYC-JE SA14 and pACYC-JE SA14-NS2A-G66A [25]. pSA14 and rG66A were grown in C6/36 cells for blood meals. The viral titer was assessed by a plaque assay titrated in BHK21 cells [40].

## Western blotting analysis and antibodies

The JEV-NS1 protein was cloned and inserted into a pET-28a (+) expression vector (Millipore) and expressed in the *Escherichia coli* BL21 DE3 strain in an insoluble form in inclusion bodies. The primers for cloning are presented in S1 Table. The protein was then solubilized in 8 M urea and purified with a TALON Purification Kit (Clontech). The protein concentrations were determined by a BCA protein assay kit (Thermo Fisher Scientific, Waltham, MA). The last 43 amino acids of NS1′ ($\Delta$NS1′$_{43aa}$) were synthesized by the Hefei SynthBiological Engineering Company. Briefly, mice were immunized with purified NS1′ protein or $\Delta$NS1′$_{43aa}$. The serum was stored at −80 °C. Cells were harvested and incubated in RIPA buffer (Beyotime) for 30 minutes on ice. Then, the samples were mixed with 5×SDS-PAGE sample buffer and boiled for 15 minutes. Equal amounts of chilled samples were separated by SDS-PAGE and transferred to PVDF membranes (Bio-Rad). The PVDF membranes were blocked with 5% nonfat milk in Tris-buffered saline and 0.1% Tween-20 (TBS-T) wash buffer and then incubated with primary antibody at room temperature for 2 h or over-night at 4 °C. The membranes were incubated with the secondary antibodies (Proteintech) at room temperature for 1 hour and washed three times for 5 minutes in TBS-T. The following antibodies were used in these experiments: anti-V5-HRP (Invitrogen) antibodies and anti-GAPDH (Proteintech) antibodies.

## JEV NS1′ protein generation in a *Drosophila* expression system

The JEV NS1′ gene was cloned and inserted into a pMT/Bip/V5-His A vector (Invitrogen) for expression in *Drosophila* S2 cells. The cloning primers are shown in S1 Table. The procedure used to generate stable cells is described in the manual of the *Drosophila* expression system. For NS1′ expression, stable S2 cells were grown in regular Schneider's Medium in a 175-cm$^2$ flask and transferred to Express Five serum-free medium (Gibco) for protein expression. After 3 days of growth, 500 µM cooper was added to the medium. For another 4 days, the supernatant was centrifuged, filtered, and purified with a TALON Purification Kit (Clontech). Protein purity was verified by SDS-PAGE and Western blotting with an anti-V5-HRP antibody (Thermo).

## Purification of infectious JEV virions

The parent JEV (pSA14) virions and rG66A virions were purified by high-speed centrifugation [16,41]. After 5 days of inoculation, the supernatant from the infected cells was collected. To remove the cell fragments, the supernatant was centrifuged at 25,000 ×$g$ and 4 °C for 20 minutes. Next, the supernatant was transferred to another centrifuge tube and centrifuged at 25,000 ×$g$ and 4 °C for 6 hours. The precipitated virions were solubilized in DMEM medium (Gibco) and kept at −80 °C.

## Plaque assay

The infected cell cultures supernatants were harvested and stored at −80 °C. To release the intracellular viral particles from cell pellets, cells were rinsed with PBS, given new DMEM, scraped and lysed by repeated freezing and thawing in dry $CO_2$ and a 37 °C

water bath. Viral particles in extracellular and intracellular were combined to measure viral load [40]. The samples were diluted and inoculated into BHK21 cells at 37 °C for 2 hours. Then, the supernatants were removed, and the cells were washed with PBS. BHK21 cells were then incubated for 4–5 days in DMEM supplemented with 2% FBS and 1% penicillin-streptomycin-amphotericin B solution (BBI). Viral titers were calculated based on visible plaques after staining with crystal violet. All the data are expressed as the means of triplicate samples.

## Viral genome quantitation by TaqMan qPCR

RNA was extracted by using a Total RNA Extraction Kit (AXYGEN), and reverse-transcription into cDNA was performed via a cDNA reverse transcription kit (Transgen). To quantify the genome burden, quantitative real-time PCR was performed on a Bio-Rad CFX96 Touch Real-Time Detection System. The primers and probes used are shown in S1 Table. Gene expression was normalized to that of the mosquito actin gene.

## Membrane blood feeding

To avoid any potential unwanted effects, the mouse blood was processed as outlined below. To separate plasma from blood cells, fresh mouse blood was centrifuged at 1,000g and 4 °C for 10 minutes. The plasma was then collected and kept at 55 °C for 60 minutes. At the same time, to remove the anticoagulant, the separated blood cells were washed with PBS. After washing 3 times, the separated blood cells were added to heat-inactivated plasma. Purified protein or antibodies, viruses, and mouse blood were mixed for feeding mosquitoes via a Hemotek system (6W1, Hemotek Limited, England). After 40 minutes, the engorged female mosquitoes were separated into new containers and kept for at least one week to analyze the infection ratios.

## Gene silencing in mosquitoes

Female mosquitoes were anesthetized on a cold tray, and 1μg/300nl of double-stranded RNA (dsRNA) was microinjected into their thoraxes. The injected mosquitoes were allowed to recover for 3days under standard rearing conditions. They were subsequently used for micro-injected. Gene silencing efficiency was assessed by qPCR. The primers used for gene detection are shown in S1 Table.

## RNA-Seq analysis of mosquitoes

Total RNA was isolated from *Culex quinquefasciatus* (the Chengdu strain) specimens following infection with rG66A virions and JEV NS1′ protein (Day 2 and Day 3). An equivalent concentration of BSA and rG66A virions served as the negative control. The samples were delivered to LC Bio Technology CO., Ltd. (Hangzhou, China) for commercial RNA-Seq services and data analysis. Clean reads were mapped to the *Culex quinquefasciatus*. Transcript database using SOAPaligner/ SOAP2 mismatches. The number of clean reads for each gene was calculated and then normalized to reads per kilobase per million reads (RPKM), which associates read numbers with gene expression levels. The $\log_2$ ratio was used to determine gene regulation. Immune genes with a $\log_2$ ratio $\leq-1$ or a $\log_2$ ratio $\geq1$ were selected for further analysis. The RNA libraries were sequenced on the Illumina Novaseq 6000 platform by LC Bio Technology CO., Ltd. (Hangzhou, China). The sequencing data associated with this project have been deposited in the NCBI Short Read Archive database (accession number: PRJNA1193468).

## Quantification and statistical analysis

Animals were randomly allocated to different groups. Mosquitoes that died before measurement were excluded from the analysis. The investigators were not blinded to the allocation during the experiments or to the outcome assessment. All experiments were performed independently at least three times. All the statistical analyses of all data were conducted by using Prism (8.0). *P* values were determined using the Mann-Whitney test. A description of the data is provided in the figure legends. Raw data for all figures are included in S2 Table.

## Results

### The G66A mutation in the NS2A of JEV eliminates the production of the NS1′ protein in mosquito vectors

The NS1′ protein relies on a pseudoknot secondary structure, with nucleotide 66 of the NS2A gene playing a pivotal role in its production [20–26]. In the NS2A coding region of the full-length cDNA clone of the JEV SA14 strain, a G66A mutation was identified (Fig 1A). This mutation, categorized as a silent point mutation, did not alter the amino acid sequence of the NS2A protein. To examine the effects of this mutation, we infected BHK21 cells and C6/36 cells with both the parental virus (pSA14) and the NS1′-deficient virus (rG66A) at an MOI of 0.1. Cell lysates were collected at 48 and 72 hours post-infection. As anticipated, in contrast to the pSA14-infected cells, the rG66A-infected BHK21 and C6/36 cells exclusively expressed the NS1 protein (Fig 1B and 1D). Based on previous research suggesting the potential secretion of NS1′ into the extracellular environment, we proceeded to investigate the presence of the NS1′ protein in the cell supernatants. The NS1′ protein was conspicuously absent from the cell supernatant of the rG66A-infected cells (Fig 1C and 1E). Quantitative analysis of the western blot, presented below the images, demonstrated that the mutant virus indeed reduces the production of the NS1′ protein. These results showed that we generated a strain lacking the NS1′ protein, which will serve as a valuable tool for further experimental investigations.

### The NS1′ protein plays a crucial role in promoting JEV replication within mosquito cell lines

According to previous studies, the NS1′ protein may play a role in the release step of RNA synthesis, as demonstrated by the reconstitution of RNA-dependent RNA polymerase (RdRp) activity [36]. Therefore, we hypothesized that the NS1′ protein did also impact RNA levels in mosquito cells. To test this hypothesis, we infected the C6/36 and Cxq cell lines with pSA14 or rG66A at an MOI of 0.1. We observed that the viral load in rG66A-infected cells was significantly lower compared to cells infected with pSA14 (Fig 2A and 2C). The results of the plaque assay for infectious viral particles were consistent with the RT-qPCR results (Fig 2B and 2D). Quantitative RT-qPCR also revealed that NS1′ did not enhance the viral RNA copy number in BHK21 cells (S1 Fig), which is practically consistent with previous research result [28].

To eliminate any potential confounding effects caused by the culture medium, we purified the pSA14 and rG66A virions from the supernatant of pSA14-infected and rG66A-infected C6/36 cells. Because the C-terminal 24 residues of E were retained to function as the signal peptide of NS1, we subcloned the $E_{24aa}$-NS1′region in pAC5.1/V5-HisA and transfected the segments into C6/36 cells to compensate for the NS1′ protein. The empty vector was used as a negative control. Then, the cells were infected with the two virions 2 days post transfection. To confirm NS1′ protein expression, Western blot analysis was performed (S2 Fig). As shown in Fig 2E, the expression of NS1′ significantly increased the viral load, implying that NS1′ has the ability to enhance viral infection in mosquito cells.

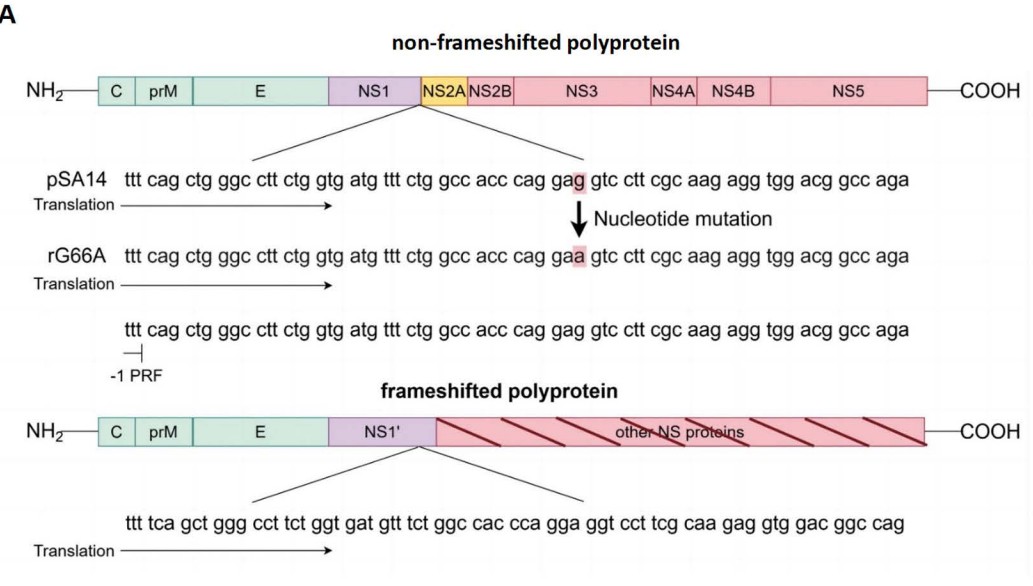

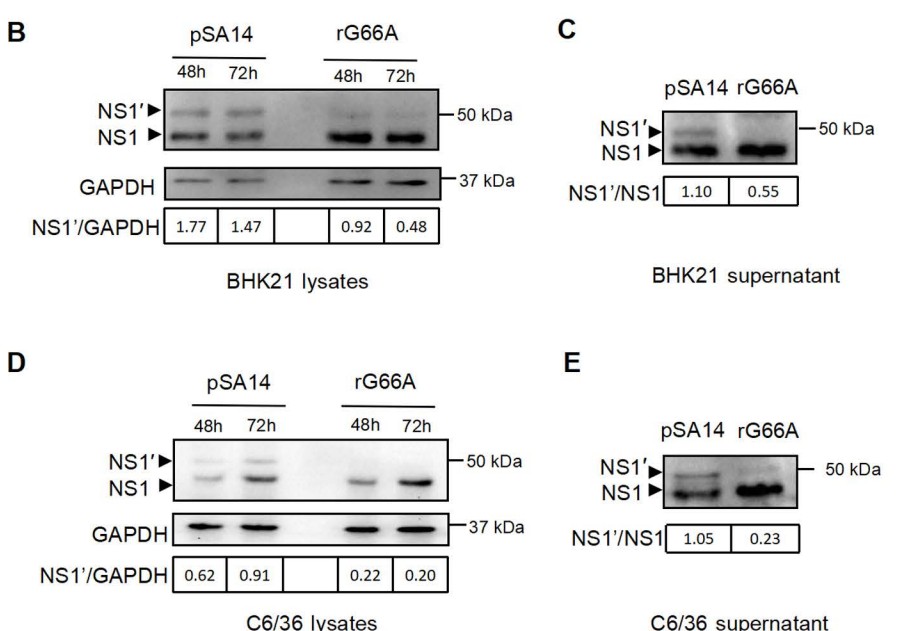

**Fig 1. The G66A mutation in the NS2A of JEV eliminates the production of the NS1′ protein in mosquito vectors.** (A) Schematic depiction of the translational path of non-PRF and PRF flavivirus polyproteins. Schematic diagram showing the processing of the flavivirus polyproteins into structural and nonstructural proteins. Numbers represent amino acid positions from the pSA14 genome. The bottom of the schematic diagram shows the site of the mutation in pSA14. (B–E) Western blot showing the protein expression of NS1 and NS1′ in infected BHK21 cells and C6/36 cells. Lysates and cell supernatants were heat denatured and analyzed by western blotting with an anti-NS1 antibody. (C and E) We infected cells with pSA14 and rG66A at an MOI of 1. At 48 hours post infection, we harvested the cell supernatant and assessed the NS1′ protein by western blotting. The intensity of bands was calculated using ImageJ software.

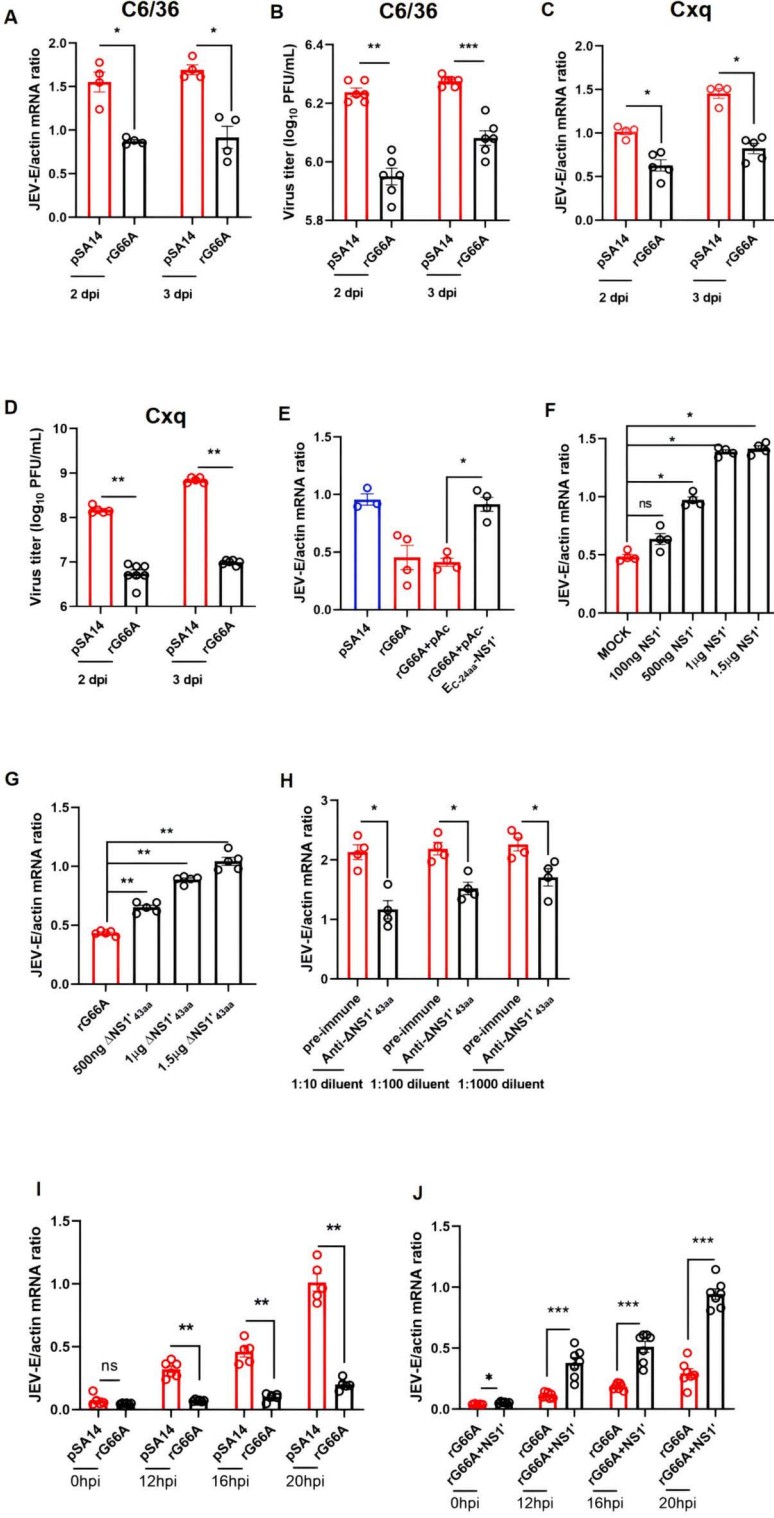

**Fig 2. The NS1′ protein plays a crucial role in promoting JEV replication within mosquito cell lines.** The mRNA levels of the E genes were determined by qPCR. (A and C) C6/36 cells and Cxq cells were infected with pSA14 or rG66A at an MOI of 0.1. The cell lysates were harvested at 2 days or 3 days after infection. (B and D) C6/36 cells and Cxq cells were infected with pSA14 or rG66A at an MOI of 0.1. The cell supernatants and cell lysates were harvested at 2 days or 3 days for assay plaque after infection. (E) C6/36 cells were transfected with empty vectors or vectors

expressing NS1′. After 2 days, the cells were infected with pSA14 or rG66A. The cells were collected after 3 days post infection. The expression of the empty vector and vectors expressing NS1′ were analyzed by western blotting using an anti-V5 antibody (S2 Fig). (F) Infectious virus particles were purified from the supernatant of infected C6/36 cells. The NS1′ protein was expressed and purified from *Drosophila* S2 cells (S3 Fig). Purified virions were added to C6/36 cells at an MOI of 0.1. (G) Infectious virus particles were purified from the supernatant of infected C6/36 cells. We synthesized the last 43 amino acids of NS1′ ($\Delta$NS1′$_{43aa}$), and the peptides were added to infected cells. The virions were added to C6/36 cells at an MOI of 0.1. (H) Serially diluted anti-$\Delta$NS1′$_{43aa}$ antisera were added to the supernatant of pSA14-infected cells. As a mock control, the same dilution of pre-immune sera was added to the cells. (I and J) Viral early replication in C6/36 cells was assessed by RT-qPCR. (K) C6/36 cells were infected with pSA14 or rG66A. Viral burden was measured at the indicated time points after infection. (L) C6/36 cells were infected with rG66A virion and NS1′ protein. Viral burden was measured at the indicated time points after infection. (A–J) The data are presented as the mean ± SEM. A nonparametric Mann-Whitney test was used for the statistical analyses. *$P < 0.05$, **$P < 0.01$, ***$P < 0.001$; n.s., not significant. The experiments were biologically repeated at least 3 times with similar results.

To further investigate the function of the NS1′ protein, we used the *Drosophila* S2 expression system to express and purify the NS1′ protein (S3 Fig). Subsequently, the purified protein was introduced into infected cells at various concentrations. The introduction of the NS1′ protein resulted in a significant increase in viral burden within cells infected with rG66A (Fig 2F). NS1′ protein contains the entire NS1 sequence, the first 9 amino acids of NS2A protein, and 43 amino acids unique to NS1′. Therefore, we introduced the last 43 amino acids of NS1′ ($\Delta$NS1′$_{43aa}$) into C6/36 cells following infection with rG66A. This led to an increase in the mRNA levels of JEV-E, thereby confirming our hypothesis that the additional 43 amino acids in NS1' can enhance viral replication in mosquitoes (Fig 2E).

To investigate the potential involvement of the NS1' protein in virus replication, we generated murine polyclonal antibodies targeting the last 43 amino acids of NS1′(anti-$\Delta$NS1′$_{43aa}$). These antibodies demonstrated the ability to recognize the full-length NS1′ protein (S4 Fig). In our experimental setup, C6/36 cells were infected with pSA14 and serially diluted antibodies were added to the culture medium. Notably, compared with the pre-immune serum, these antibodies significantly reduced the JEV E mRNA level in C6/36 cells (Fig 2H).

To elucidate the regulatory role of NS1′ in the propagation of JEV in mosquito cell lines, we initially conducted virus attachment assay (S5A and S5B Fig). The experimental data indicated that NS1' protein does not facilitate virus attachment. Subsequently, virus internalization assays were conducted, demonstrating that NS1' protein dose not enhance virus internalization (S5C and S5D Fig). To further explore the function of the NS1' protein in viral replication within C6/36 cell line, we collected cells at 0, 12, 16, and 20 hours post-infection. The results revealed that the NS1' protein significantly enhanced viral replication in C6/36 cells (Fig 2I and 2J). These findings suggested that NS1′ can enhance JEV replication in mosquito cells, necessitating further comprehensive investigations.

## The NS1′ protein enhances JEV infection in *Culex quinquefasciatus*

*Culex* mosquitoes are the main vector for JEV transmission. To further explore the role of NS1′ in JEV infection vectors, equal numbers of viral particles were initially injected into mosquitoes *via* microinjection. After 3 or 6 days, the viral loads were assessed. Consistent with the findings observed in cell lines, the mRNA levels in mosquitoes injected with rG66A were lower than those in mosquitoes injected with pSA14 (Fig 3A). Subsequently, the mosquitoes were also fed pSA14 or rG66 through an *in vitro* blood feeding system. A mixture containing mouse blood (50% vol/vol) and supernatant from pSA14- or rG66A-infected cells (50% vol/vol) was used to feed *Culex quinquefasciatus* (Fig 3B). Compared to those fed pSA14 the rG66A- fed mosquitoes had lower infection ratios (Fig 3C and 3D).

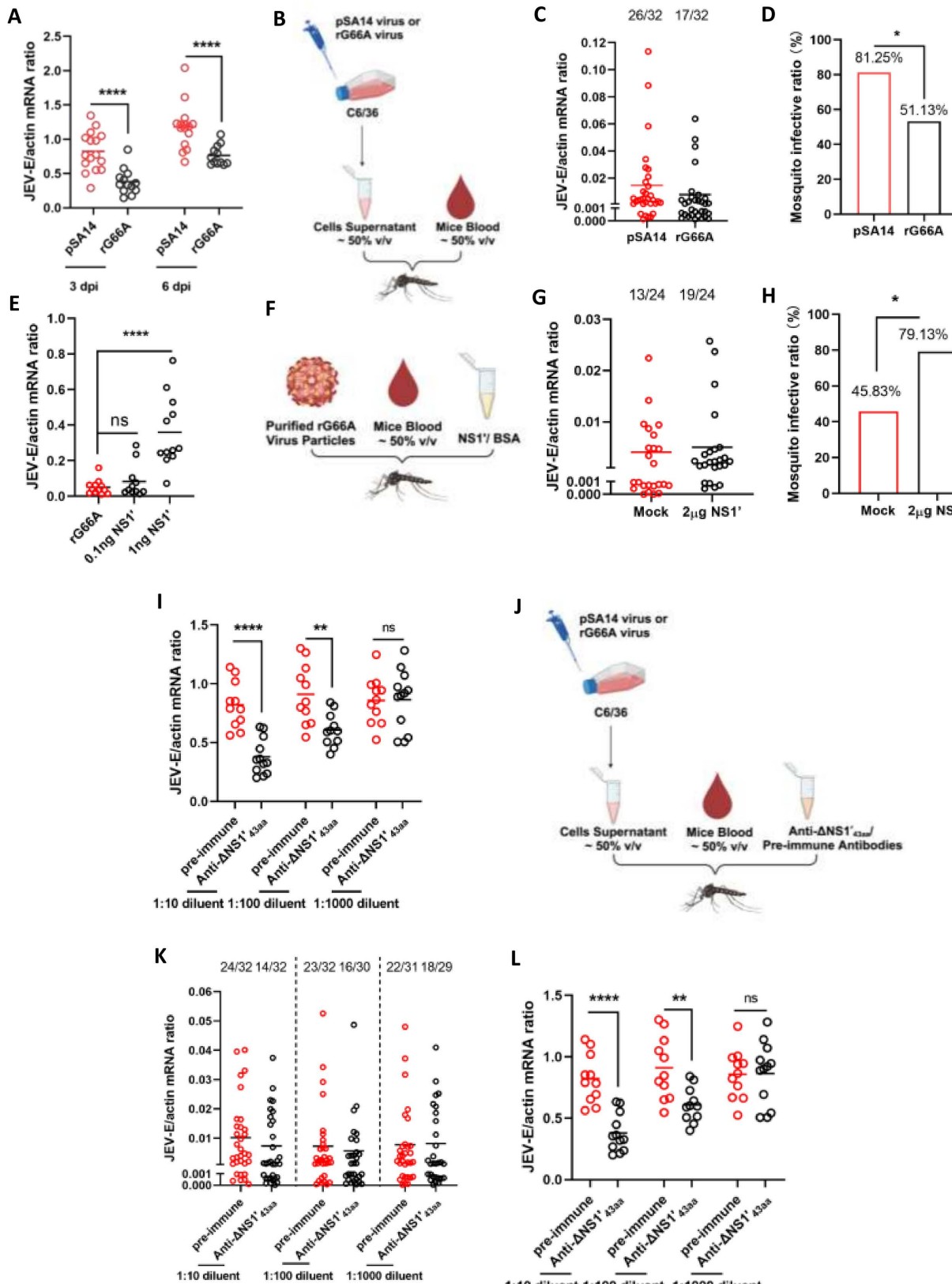

**Fig 3. The NS1′ protein enhances JEV infection in *Culex quinquefasciatus*.** (A–D) The rG66A mutation results in reduced replication in *Culex quinquefasciatus* mosquitoes. (A) pSA14/rG66A was microinjected into mosquitoes at the same number of pfu/injection. The mRNA levels of the virus

in the mosquitoes were measured *via* RT-qPCR. (B–D) Supernatant from pSA14/rG66A-infected cells and fresh mouse blood were mixed for in vitro membrane feeding of *Culex quinquefasciatus* mosquitoes. (B) Created in BioRender. Yang, X. (2024) https://BioRender.com/i46s092. (E–H) The presence of NS1′ increased rG66A replication and infection in mosquitoes. (E) The NS1′ protein was expressed in and purified from *Drosophila* S2 cells. The NS1′ protein incubated with purified rG66A virions was microinjected into mosquitoes. (F–H) Purified rG66A virus particles and purified NS1′ protein were mixed with mouse blood to feed mosquitoes. (F) Created in BioRender. Yang, X. (2024) https://BioRender.com/k10t669. (I–L) Immunoblockade of NS1′ in the supernatant of infected C6/36 cells reduced JEV replication and infection in mosquitoes. (I) A mixture of serially diluted anti-ΔNS1′$_{43aa}$ antisera and supernatant from pSA14-infected C6/36 cells was microinjected into mosquitoes. (J–K) Serially diluted anti-ΔNS1′$_{43aa}$ was mixed with supernatant from pSA14-infected C6/36 cells and fresh mouse blood for in vitro membrane feeding of *Culex quinquefasciatus*. As a mock control, mosquitoes were fed the same dilution of pre-immune sera. (J) Created in BioRender. Yang, X. (2024) https://BioRender.com/i23u307. (A, E, and I) Mosquito samples were collected at 5 days after microinjection. (B–D, F–H, J–L) Mosquito infectivity was determined by TaqMan qPCR at 8 days after the blood meal. (C, G, and K) The number of infected mosquitoes relative to the total number of mosquitoes is shown at the top of each column. Each dot represents a mosquito. (D, H, and L) The data are represented as the percentage of injected mosquito. Differences in the mosquito infection ratio were compared using Fisher's exact test. (A–E) The data are presented as the mean ± SEM. A nonparametric Mann-Whitney test was used for the statistical analyses. *$P < 0.05$, **$P < 0.01$. The experiments were biologically repeated at least 2 times with similar results.

Subsequently, rG66A mixed with various amounts of purified NS1′ protein was used to infect *Culex quinquefasciatus via* thoracic microinjection. A noteworthy increase in the viral load was observed when the virus was mixed with 1ng of the NS1′ protein (Fig 3E). Next, mouse blood was incubated with purified rG66A particles and purified NS1′ protein and then orally fed to the mosquitoes (Fig 3F). In line with previous findings, the inclusion of the NS1′ protein resulted in a significant increase in the mosquito infection ratio (Fig 3G and 3H).

To facilitate a more extensive investigation, murine polyclonal antibodies against the last 43 amino acids of NS1′ (anti-ΔNS1′$_{43a}$) were subjected to serial dilution and subsequently injected into mosquitoes. Upon injection of anti-ΔNS1′$_{43aa}$, a noticeable decrease in the mRNA expression of JEV-E was observed in comparison with the injection of pre-immune serum (Fig 3I). Then, we conducted an in vitro blood feeding experiment on mosquitoes using a combination of mouse blood, the supernatant of pSA14-infected C6/36 cells, and serially diluted antibodies (Fig 3J). Similar outcomes were observed when mosquitoes were fed anti-ΔNS1′$_{43aa}$ (Fig 3K and 3L). These findings provided evidence that the presence of NS1′ facilitates JEV infection in mosquitoes. Therefore, this research provides substantial confirmation that the NS1′ protein plays a crucial role in promoting virus replication and mosquito infection.

## The JEV NS1′ protein enhances viral infection by suppressing the expression of AMPs through the toll signaling pathway in *Culex quinquefasciatu*s mosquitoes

We next focused on elucidating the mechanism by which NS1′ facilitates JEV infection of JEV in mosquitoes. Mosquitoes that were injected with purified JEV NS1′ protein and rG66A virion were sacrificed for RNA-Seq analysis. Based on the volcano plot analysis, the expression levels of the *Toll*, *REL1A*, *REL1B*, *IMD*, and *REL2* genes exhibited significant alterations at 48 and 72 hours post-microinjection (Fig 4A and 4B). Furthermore, the heat map analysis of the downregulated genes indicated that the inhibitory effect of the NS1′ protein on Toll signaling pathway and AMPs-related genes was more pronounced 72 hours after mosquito infection (Fig 4C). Mosquitoes utilize innate immune antiviral mechanisms such as reactive oxygen species (ROS) production, RNAi, and immune signals to restrict viral spread. The mRNA levels of the majority of genes within the RNAi, JAK-STAT, and IMD pathways exhibited no significant differences between the NS1′ protein and control groups (gene upregulation or downregulation less than 2-fold). Conversely, the mRNA expression levels of genes associated with the Toll pathway and several AMPs were downregulated in the NS1′-treated mosquitoes (Fig 4A–C). Subsequently, the changes in the expression of these immune-related genes at 48h

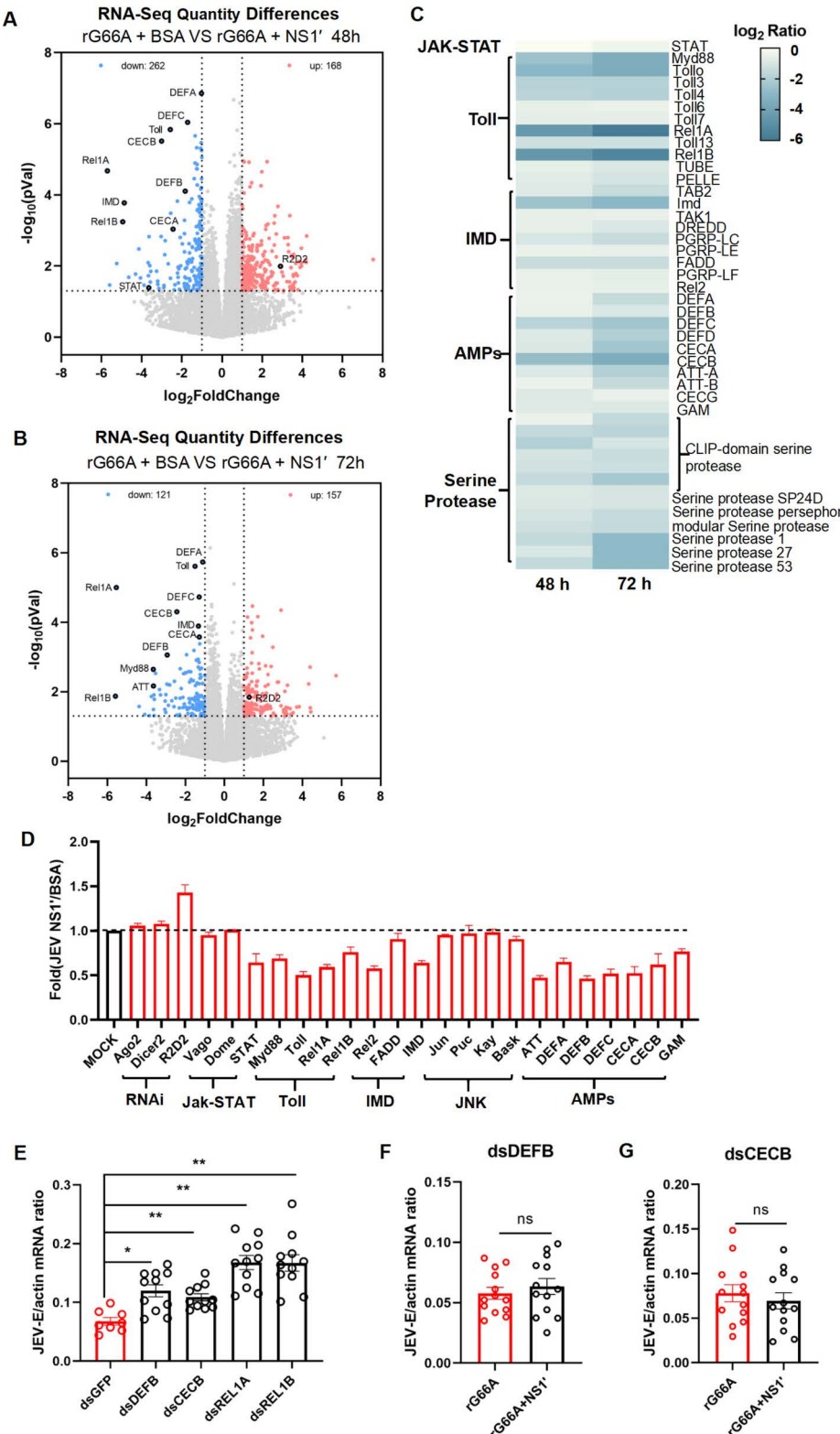

**Fig 4. The JEV NS1′ protein enhances viral infection by suppressing the expression of AMPs through the Toll signaling pathway in *Culex quinquefasciatus* mosquitoes.** (A–C) Regulation of immune-related genes in *Culex quinquefasciatus* mosquitoes. Mosquitoes that acquired rG66A virions at 10 MOI and 2 μg of purified JEV NS1′ or BSA

*via* microinjection were collected for RNA-Seq. (A and B) Volcano map of transcriptome. Toll signaling pathway and AMPs were provided. $Log_2$FC<−1 or >1, $P$ < 0.05. (A) The samples were collected at 48 hours post micro-injection. (B) The samples were collected at 72 hours post microinjected. (C) A heat map that shows the expression of immune pathway-related genes that were downregulated at both 48 hours and 72 hours after microinjection. Immune related genes were clustered according to immune pathways and factors. (D) Validation of immune-related gene regulation. The expression levels of the immune-related genes were assessed by qPCR at 72 hours post micro-injection. Gene regulation is represented as the mRNA ration between NS1′ injected and BSA injected in rG66A infected mosquitoes. The primers used are listed in S1 Table. (E–G) Mosquito infectivity was determined by TaqMan qPCR at 3 days post microinjected. The expression levels of the immune-related genes were assessed by qPCR at 3 days post micro-injection. (E) Genetic suppression of the Toll pathway and AMPs enhanced rG66A infection in mosquitoes. GFP dsRNA served as a negative control. (F and G) After 3 days of dsRNA treatment, the mosquitoes were microinjected with rG66A and NS1′. rG66A and BSA served as control. The primers used are listed in S1 Table. The data are presented as the mean ± SEM. The nonparametric Mann-Whitney test was used for statistical analysis. **$P$ < 0.01; n.s., not significant. (A–G) The experiments were biologically repeated at least 3 times with similar results.

and 72h were validated through qPCR. Specifically, the mRNA abundance of three DEFs and two CECs was reduced in the NS1′-treated mosquitoes, confirming the accuracy of the mRNA sequencing data (Figs 4D and S6).

To elucidate the roles of NS1′ in JEV infection, we silenced the genes *DEFA*, *DEFB*, *REL1A*, and *REL1B* in *Culex quinquefasciatus* and evaluated the effect on viral burden 3 days after pSA14 infection. The expression levels of these genes were significantly reduced following dsRNA treatment (S7 Fig). In comparison to the *GFP* dsRNA-inoculated control group, the knockdown of *DEFA*, *CECB*, *REL1A,* and *REL1B* genes resulted in a significant increase in JEV burden in the mosquitoes. Notably, the suppression of *REL1A* and *REL1B* led to a twofold enhancement in viral load (Fig 4E). Furthermore, the suppression of *DEFA* and *CECB* compromised the NS1' protein's influence on viral infection in mosquitoes (Figs 4F, 4G, and S8).

These findings suggested that JEV NS1′ suppresses AMPs *via* the Toll pathway to facilitate viral infection. These data indicated that NS1′ facilitates JEV infection by modulating mosquito antiviral mechanisms.

## Discussion

The significance of the NS1 protein in flaviviruses has been recognized for its crucial roles in virus replication, immune modulation, and transmission to vectors [2,3,16]. The larger NS1-related protein, NS1′, an additional form of NS1 protein with 52 amino acids extension at the C-terminal region which includes the N-terminal 9aa of NS2A and 43aa after the frameshift, is frequently observed during infections with the JE serogroup of flaviviruses. Various investigations have indicated that NS1' contributes to the suppression of innate immunity and the enhancement of neuroinvasiveness and neurovirulence in mammals [16–19]. Nevertheless, there has been limited research conducted on the interaction between the NS1' protein and its primary vector, *Culex* mosquitoes. Previous research has demonstrated that the NS1' protein of WNV can enhance dissemination from the midgut in *Culex annulirostris* mosquitoes [21]. The present study revealed that the NS1' protein enhanced JEV infection in both mosquito cell lines and mosquito vectors. Subsequent investigations indicated that the NS1' protein inhibited the key components of the Toll pathways, which are responsible for regulating AMPs [33,40,42,43]. Consequently, the expression of AMPs was suppressed by the NS1' protein, thereby promoting JEV infection in *Culex*. In contrast to the NS1 protein, which inhibits ROS and the JAK-STAT signaling pathway in mosquitoes, the NS1' protein suppresses components of the Toll signaling pathway, thereby inhibiting the expression of AMPs. Because the 43aa are unique to NS1′, we synthesized a peptide fragment consisting of the C-terminal 43aa of the NS1'protein. Subsequently we generated an antibody specifically targeting the last 43aa to selectively block its

function. In our experiment, we introduced the 43aa peptide into cells and observed an enhancement in viral replication. Moreover, inhibiting its function with an antibody targeting the 43aa sequence resulted in reduced viral proliferation. This suggested that the regulation of the Toll signaling pathway is not mediated by the conserved sequence in NS1 but rather by the unique 43aa sequence. However, further investigation is required to elucidate the mechanisms by which these 43aa influence those pathways and the extent to which this strategy is employed by other JE serological viruses need to be further revealed.

The functions of mosquito-borne flavivirus nonstructural proteins in mammals are well defined, with some functions being conserved in mosquito vectors. In addition to these conserved roles, mosquito-borne flaviviruses are able to establish persistent infections in mosquitoes through interactions between their nonstructural proteins and the innate immune system. Previous studies have demonstrated that the NS5 and NS1 of WNV facilitate the proteasomal degradation of *Culex quinquefasciatus* STAT by upregulating the E3 ubiquitin ligase cullin4 [36]. Furthermore, the capsid proteins of DENV, WNV, ZIKV and YFV have been shown to inhibit the cleavage of dsRNA by Dicer-2 [37,38]. Our findings indicated that the NS1′ protein plays a role in enhancing JEV infection in mosquito vectors, contributing to our understanding of the function of NS1′ protein and the molecular mechanisms involved in viral-host interactions.

The live attenuated JEV vaccine, specifically JEVSA14-14-2, has demonstrated a high level of safety. Pigs inoculated with JEVSA14-14-2 exhibit transient infection, which is potentially linked to the abolishment of the NS1′ protein formation [22]. Compared to the SA14 strain, the SA14-14-2 strain showed limited infectivity in mosquito vectors after oral feeding, with replication occurring at a lower level following intrathoracic inoculation [39]. Our experimental findings align with prior research, which indicated that abolishing production of the NS1′ protein through a single mutation partially hindered the virus to infect *Culex* vectors. This finding suggested that the NS1′ protein may play a role in enhancing viral infection in mosquitoes. However, the rG66A virus retains the ability to infect mosquitoes by blood feeding even after the NS1′ protein was abolished, indicating the presence of other factors that contribute to the differences in vector infectivity between the two strains. Consequently, additional long-term studies are warranted to further investigate these findings.

Various innate immune responses are crucial for combating viral infections in mosquitoes. The commensal microbiota of the mosquito gut plays a complex role in determining the vector competence for arboviruses [44]. Research has demonstrated that the Toll pathway plays a significant role in exerting antiviral effects in mosquitoes, particularly in response to DENV infection [33,40]. This pathway is activated upon DENV infection, leading to the upregulation of various immune components such as *defensins* and *cecropins*, which have strong antiviral activity and play a crucial role in blocking DENV replication in *Aedes aegypti* [40,41,45–48]. Furthermore, the Toll pathway is recognized for its ability to trigger antiviral responses against DENV infection in mosquitoes [47]. AMPs are evolutionarily conserved elements that are governed by innate immune pathways and serve as potent antiviral effectors. In insects, AMPs have been shown to effectively combat bacteria, fungi, and viruses. A peptide similar to CEC, induced by DENV, has been identified as a limiting factor for the virus in the salivary glands of *A.aegypti* [49]. Various AMPs belonging to the DEF family have exhibited anti-DENV properties in *A.aegypti* [50]. Nonetheless, the involvement of the DEF family is intricate, as prior research has shown that DEFA directly binds to the JEV E protein, aiding in the adsorption of JEV in *Culex* mosquitoes [51]. A majority of AMPs exhibit antiviral activity primarily by inhibiting viral attachment and membrane fusion, destroying the viral envelope, or suppressing viral replication [52]. Melittin and cecropin,

two extensively studied antimicrobial peptides (AMPs), demonstrate antiviral properties by suppressing the expression of HIV-related genes [53]. Similarly, ATT and diptericin have been shown to exhibit antiviral activity by regulating viral RNA synthesis during Sindbis virus (SINV) infection in *Drosophila* [54]. In mosquitoes, several AMPs, including DEFA, DEFE, and CECE, may be upregulated following flavivirus infection [29]. In *Aedes aegypti*, the AaSR-C/AaMCR pathway induces the expression of AMPs to counteract dengue virus (DENV) infection [37]. In a virus host arms race, viruses have evolved intricate strategies to effectively infect their host. Several flavivirus proteins, specifically the nonstructural proteins, are involved in facilitating viral immune evasion in mammals [2,13,16,24,55]. The Toll pathway is crucial for restricting flavivirus infection in mosquitoes. Correspondingly, inhibition of the Toll pathway has been demonstrated to enhance DENV infection in *A. aegypti* [40]. The potential interference of viral proteins with antiviral mechanisms requires further investigation. Our research indicated that JEV relies on the NS1′ protein to evade the Toll pathway. Furthermore, we observed that the JEV NS1′ protein, which is distinct from NS1, inhibited the expression of key components in the Toll pathway, subsequently leading to the downregulation of the expression of AMPs, including the Toll pathway marker genes, such as *ATT*, *DEFA, DEFB, DEFC* and *CECB*. The mechanism through which the NS1' protein inhibits antimicrobial peptides (AMPs) *via* the Toll pathway requires further investigation. The NS1' protein can be secreted extracellularly and may interact with extracellular proteins associated with the Toll pathway, such as leucine-rich repeat-containing proteins and clip-domain serine proteases. Additionally, the NS1' protein can be internalized into mosquito cells, potentially affecting the regulation of downstream genes within the Toll pathway [44]. Overall, our study provides insight into the essential role and potential mechanisms of the NS1′ protein in evading the immune response and promoting mosquito infection. Nevertheless, further research is needed to clarify the relationship between the NS1′ protein and the Toll signaling pathway.

In conclusion, our findings demonstrate the pivotal role of the NS1′ protein in promoting virus infection and replication in mosquitoes. Mechanistically, we observed that the NS1′ protein suppresses the expression of several AMPs by inhibiting Toll. Furthermore, we have initiated preliminary research to elucidate the underlying mechanisms of this phenomenon. Overall, JEV is a notable zoonotic disease that presents a substantial risk to human health and economic stability. Our research revealed a new approach by which JEV evades the innate immune response of mosquitoes, with the goal of suggesting improved strategies for prevention and control.

## Supporting information

**S1 Fig. The mRNA levels of E genes in pSA14 or rG66A infected BHK21 cell.** BHK21 cells were infected with pSA14 or rG66A at an MOI of 0.1. The cell lysates were harvested at 1day or 2days after infection. The mRNA levels of E genes were determined by qPCR. The data are presented as the mean ± SEM. A nonparametric Mann-Whitney test was used for the statistical analyses. *P < 0.05, **P < 0.01. The experiments were biologically repeated at least 3 times with similar results.
(TIF)

**S2 Fig**. **The expression of empty vector and vectors expressing NS1′ were analyzed by western blotting using anti-V5.** The NS1′ genes were cloned into a pAC-V5-HisA vector. C6/36 cells were transfected with empty vectors or vectors expressing NS1′. Expression of empty vector and vectors expressing NS1′ were analyzed by western blotting using anti-V5.
(TIF)

**S3 Fig. Purification of JEV NS1′ protein from *Drosophila* S2 expression system.** The NS1′ genes were cloned into pMT/BiP/V5-HisA vector. And NS1′ protein was expressed and purified on Econo Fit Nuvia IMAC column (Left panel). Protein expression was evaluated using an anti-V5-HRP mAb (Right panel).
(TIF)

**S4 Fig. Generation of murine polyclonal antibodies against the last 43 amnio acid of NS1′ (ΔNS1′$_{43aa}$).** The last 43 amnio acid of NS1′ (ΔNS1′$_{43aa}$) were synthesized in Hefei SynthBiological Engineering Company. The mice were immunized with 8ug ΔNS1′$_{43aa}$ each. And the The mice serum was stored at −80 °C. The JEV infected cells lysates weas loaded into each lane. The same samples probed by mouse pre-immune antibody served as a negative control.
(TIF)

**S5 Fig. NS1 protein affects JEV replication but not viral adsorption and internalization in C6/36 cells.** (A and B) Viral attachment was assessed by RT-qPCR. (A) C6/36 cells were incubated with JEV at an MOI of 5 for 1 h at 4 °C. The unbound virus was removed by washing with PBS. After washing with PBS, the cells were assessed by RT-qPCR. (B) C6/36 cells were incubated with rG66A viron and 1μg NS1' protein at an MOI of 5 for 1 h at 4 °C. (C and D) Viral internalization into cells was assessed by RT-qPCR. (C) For the internalization assay, after JEV adsorption at 4 °C for 1 h, the cells were washed with PBS and subsequently incubated to 30 °C for 1 h to allow virion internalization. After washing with PBS, the cells were treated with proteinase K to remove noninternalized virions. (D) C6/36 cells were infected with rG66A virus and 1 μg NS1' protein at an MOI of 5. (A–D) The data are presented as the mean ± SEM. The nonparametric Mann-Whitney test was used for statistical analysis. \*\**P* < 0.01; n.s., not significant.
(TIF)

**S6 Fig. Validation of immune-related gene regulation.** The expression levels of the immune-related genes were assessed by qPCR at 48 hours post micro-injection. Gene regulation is represented as the mRNA ration between NS1′ injected and BSA injected in rG66A infected mosquitoes. The primers used are listed in S1 Table. The data are presented as the mean ± SEM. The nonparametric Mann-Whitney test was used for statistical analysis. \*\**P* < 0.01; n.s., not significant.
(TIF)

**S7 Fig. The gene silencing efficiency in *Culex quinquefasciatus* was detected by RT-qPCR.** (A–D) Total RNA was extracted, gene levels were detected by RT-qPCR. Gene expression was normalized to the *Culex quinquefasciatus* actin gene. Data are represented as mean ± SEM. in each group and analyzed using the nonparametric Mann Whitney test. \*\**P* < 0.01, \*\*\*\**P* < 0.0001.
(TIF)

**S8 Fig. The gene silencing efficiency in *Culex quinquefasciatus* was detected by RT-qPCR.** (A and B) Total RNA was extracted, gene levels were detected by RT-qPCR. Gene expression was normalized to the *Culex quinquefasciatus* actin gene. Data are represented as mean ± SEM. in each group and analyzed using the nonparametric Mann Whitney test. \*\**P* < 0.01, \*\*\*\**P* < 0.0001.
(TIF)

**S1 Table. Primers and probe used for qPCR and genes cloning.**
(XLSX)

**S2 Table. Raw data for all figures.**
(XLSX)

## Acknowledgments

We thank Yajie Hu from the Sichuan Center for Disease Control and Prevention for donating the *Culex quinquefasciatus* Chengdu strain.

## Author contributions

**Conceptualization:** Senyan Du.

**Data curation:** Yuwei Liu, Yutian Huang, Yi-fei Lang.

**Funding acquisition:** Senyan Du.

**Investigation:** Yuwei Liu, Yutian Huang, Yi-fei Lang.

**Methodology:** Yuwei Liu, Yutian Huang, Yi-fei Lang.

**Resources:** Yi He, Changhao Xu, Xi Zhu, Bowen Li, Yajie Hu.

**Software:** Ruidong Li, Chang Miao.

**Supervision:** Rui Wu, Qin Zhao, Yiping Wen, Xiaobo Huang, Qi-gui Yan, Shan Zhao, Yiping Wang, San-jie Cao.

**Validation:** Yuwei Liu, Yutian Huang, Yi-fei Lang.

**Visualization:** Yuwei Liu, Yutian Huang, Yi-fei Lang.

**Writing – original draft:** Senyan Du.

**Writing – review & editing:** Senyan Du.

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
