## [Decision Letter · Decision Letter 0]

5 Oct 2024

Dear Miss 杜,

Thank you very much for submitting your manuscript "The Japanese encephalitis virus NS1′ protein facilitates virus infection in mosquitoes" for consideration at PLOS Neglected Tropical Diseases. As with all papers reviewed by the journal, your manuscript was reviewed by members of the editorial board and by several independent reviewers. In light of the reviews (below this email), we would like to invite the resubmission of a significantly-revised version that takes into account the reviewers' comments.

All 3 reviewers found your work interesting although they raised a number of issues that need to be addressed. In particular, I would ask that you focus on showing the functionality of the NS1' protein more clearly by including experimental data as suggested by Reviewers 1 and 3. Without such experimental data, the drawn conclusions cannot be fully supported.

We cannot make any decision about publication until we have seen the revised manuscript and your response to the reviewers' comments. Your revised manuscript is also likely to be sent to reviewers for further evaluation.

Sincerely,

Eng Eong Ooi

Guest Editor

Andrea Marzi

Section Editor

All 3 reviewers found your work interesting although they raised a number of issues that need to be addressed. In particular, I would ask that you focus on showing the functionality of the NS1' protein more clearly by including experimental data as suggested by Reviewers 1 and 3. Without such experimental data, the drawn conclusions cannot be fully supported.

Reviewer's Responses to Questions

**Key Review Criteria Required for Acceptance?**

**Methods**

-Are the objectives of the study clearly articulated with a clear testable hypothesis stated?

-Is the study design appropriate to address the stated objectives?

-Is the population clearly described and appropriate for the hypothesis being tested?

-Is the sample size sufficient to ensure adequate power to address the hypothesis being tested?

-Were correct statistical analysis used to support conclusions?

-Are there concerns about ethical or regulatory requirements being met?

Reviewer #1: Experiments using mosquitoes are performed very well. Results are all consistent.

There are several issues in experiments using cell lines.

Reviewer #2: (No Response)

Reviewer #3: Methods and objectives were described clearly. No issues with sample size or ethics.

**Results**

-Does the analysis presented match the analysis plan?

-Are the results clearly and completely presented?

-Are the figures (Tables, Images) of sufficient quality for clarity?

Reviewer #1: All data is not enough to conclude authors' conclusion. More experimental data is needed.

Reviewer #2: (No Response)

Reviewer #3: Results and figures presented clearly. However, there are some improvements that should be considered. See comments below for more details.

**Conclusions**

-Are the conclusions supported by the data presented?

-Are the limitations of analysis clearly described?

-Do the authors discuss how these data can be helpful to advance our understanding of the topic under study?

-Is public health relevance addressed?

Reviewer #1: Conclusions authors claimed in this study are interesting and contribute in this research fields.

Reviewer #2: (No Response)

Reviewer #3: Effects of rG66A on virus infection is well illustrated. However, the RNAseq may need further analysis to confirm that the mechanisms are indeed through suppression of antimicrobial peptides.

**Editorial and Data Presentation Modifications?**

Reviewer #1: 1. Line 232 needs references or evidence.

2. Figure 1B–E needs quantification data.

3. In Figure 2, the authors performed only qPCR. The entire phenotype of NS1′ protein for viral replication is not very apparent, since all the data were significantly but not so high (about 2–3 fold change in RNA levels). How NS1′ regulates JEV propagation in mosquito cell lines remains unclear. The ratios of viral attachment on cell membrane and viral titers should be quantified.

4. Authors added NS1′ protein into virus infected cells. Authors should confirm NS1′ was found intracellular.

5. In Line 386, explain how NS1′ suppresses AMPs via the Toll pathway? If the AMP pathway is affected by NS1′, how did AMP impair viral RNA as shown in Figure 2?

6. In Figure 4, the authors showed that NS1′ administration downregulated the mRNA expression levels of genes associated with the Toll pathway and several antimicrobial peptides (AMPs) in mosquitoes, using the RNA-seq and qPCR. However, the effect of these AMPs on JEV replication is unclear. The author should measure the expression levels of AMPs and genes associated with the Toll pathway in pSA14- or rG66A-infected cells and mosquitoes. Moreover, NS1′ protein administration cannot ameliorate the effect of NS1 region, because previous report showed that NS1′ shares a similar cellular distribution with NS1(19). Since the authors demonstrated that the peptides of ΔNS1′44aa enhanced viral replication in mosquito cells (Figure 2E), I recommend the use of ΔNS1′44aa for microinjection in Figure 4.

7. Why did the author use introduced ΔNS1′44aa but not ΔNS1′52aa (the last 52 unique amino acid of NS1′ protein) as reported previously (PMID: 31996459)?

8. Previous report (PMID: 24443559) showed that NS1′ protein efficiently co-localizes with NS5 protein and increases viral RNA levels in avian cells. Co-immunoprecipitation assays of NS1′ and NS5 in BHK21 and mosquitoes cell line are needed.

9. All data lacks how P values were determined by Prism software.

Reviewer #2: (No Response)

Reviewer #3: Data presentation is generally good, but more clarity required for mechanisms of action.

**Summary and General Comments**

Reviewer #1: Liu et al. investigated the role of NS1′ in JEV propagation in mosquito cells. They demonstrated that the JEV NS1′ protein facilitates virus replication in mosquito cells and enhances JEV infection in Culex mosquitoes. Moreover, they claimed that the NS1′ protein suppresses the expression of several antimicrobial peptides (AMPs) by inhibiting the Toll pathway. This study is interesting because it provides new insights into JEV transmission mechanisms through the viral non-structural protein. Although I agree that the NS1′ protein promotes virus infection and replication in mosquitoes, my main concern is that the figures and results presented here do not fully corroborate the authors' findings, particularly how NS1′ regulates the JEV life cycle (attachment, entry, RNA replication, assembly, or release). I hope addressing the following comments will help improve the manuscript.

Reviewer #2: The authors evaluated the function of NS1’ in JEV infection in mosquitoes. The hypothesis was based on NS1’ pro-viral function in mosquitoes previously demonstrated for WNV. The study starts with the production of a NS1’ deficient JEV infectious clone. The authors then show that NS1’-deficient virus replicates less in mosquito cells and that the reduced infection is compensated by NS1’ complementation through either plasmid expression and protein supplementation. The last 44 aa of NS1’ are sufficient to reproduce the full-length NS1’ enhancing effect and the pro-viral effect is inhibited by anti-NS1’ antibody. They support NS1’ pro-viral effect in vivo by quantifying an increased infection in mosquitoes after infection by injection and oral feedings. Using complementation and immuno inhibition approaches as in cells, they confirm that NS1’ is responsible for the infection enhancement. Finally, they conducted RNA seq in NS1’ injected-mosquitoes and found downregulation of AMPs and Toll, suggesting a mechanism. Overall, the study strongly demonstrates that NS1’ enhances infection in mosquitoes, although the discovery of NS1’ function in mosquitoes was previously discovered for WNV, limiting the novelty. The suggestion about the associated mechanism are interesting less well supported and would require further discussion. Specifically, the authors should discuss whether the downregulation of AMPs and Toll components can be caused by the NS1 sequence that is conserved in NS1’.

I recommend publication after minor changes. There should also be a significant English editing.

Short title is not clearly written.

Figures:

Fig. 1: It should be noted that there remain a band for NS1’, although I agree it is clearly reduced.

Fig.4. the color code is confusing. Generally green means it is increased. Please reverse the color code.

Add the gene names in the heatmap.

Minor comments:

l. 48: a critical

l. 57: the sentence is not clear. Correct “and” for “which”

l. 62: “child as the primary host”… the formulation is suprising. Children are not a primary host but maybe the most affected by the infection.

l. 75: replace solitary with unique.

l. 99: remove “in”

l. 111: add reference for the antiviral activity of AMPs

l. 132. Add the reference for the cell line.

l. 139-140: if not providing a brief description of the methods, refer to references.

l. 151: minon ?

l. 161: subtitle is badly written

l. 173-179: centrifugation at 25 000 g should not precipitate virions… did the authors make an error in centrifugation speed?

l. 189: provide the primer sequences

l. 230-245: this whole section should be thoroughly improved for English.

l. 266: replace “couldn’t” with “did not”

l. 286: precise what hypothesis is supported here

l. 330: not clear how mosquitoes were infected here: injection or oral feeding? Please precise.

l. 373: mosquitoes were dissected but what tissues were analysed? Also, precise

l. 377: JNK is also anti-viral pathway. Add the regulation of the components of this pathway: c-Jun, Kay, Puckered, basket.

l. 416: there is one to many “serological”l. 432: intrathoracic

l. 442: references are missing

Reviewer #3: In this study, the authors explore the potential role of the NS1 protein (or more precisely NS1') in mosquito infection, by leveraging on NS1'-defective virus (rG66A). While the results are interesting, there are some concerns that should be addressed, particularly the RNAseq data. The details are as follows:

1. While it is interesting that rG66A resulted in reduced NS1' expression but not NS1 expression, the authors did not discuss nor clarify why the single nucleotide substitution resulted in reduced NS1' expression. This explanation is necessary as all subsequent studies are related to observations found in rG66A.

2. Figure 2E. Why did the authors transfect the last NS1'44aa? While I understand that the 52aa extension of the C-terminus makes NS1' unique from NS1, does the 44aa encode for the active site of NS1' activity? This should be elaborated and discussed.

3. Figure 3D, H and L are missing data points. Does this mean that the experiments are only performed once?

4. Figure 3C and G. The splitting of the y-axis (especially Figure 3C with 2 split axis) makes the figure hard to interpret, as it is difficult to visualise the distribution of the data. The authors can consider log-transformation if the differences are hard to visualise.

5. Figure 5. Besides performing comparisons of the purified NS1' vs BSA under uninfected conditions, the authors should also do purified NS1' vs BSA under infection conditions. This is because the innate immune responses (JAK-STAT, Toll and IMD) are triggered during virus infection, so this will allow the authors to understand if NS1' affects the induction of these responses which has been shown to impact virus infection outcome.

6. Figure 5A. Red and green is very hard to visualise, especially that the gradient of colours span from -3 to 1. Furthermore, the colours are not colour-blind friendly, and do not allow readers to understand the variation of the data. The authors should consider plotting volcano plots, which allows readers to quickly visualise the effect size and significance of the differences.

6. Figure 5B. It is unclear why the dotted lines are drawn at the 0.5 fold-change. Again, plotting volcano plots will help clarify the differences better.

7. Finally, as the authors have done RNAseq, were there other hits that could be interesting? Could there be other mechanisms involved other than AMPs? If it is only AMPs, would altering the expression of AMPs affect the infection outcome? The authors should report these findings and discuss what they have found.

8. Authors should also deposit their RNAseq data in GEO or ArrayExpress so others can access the data.

PLOS authors have the option to publish the peer review history of their article (what does this mean? ). If published, this will include your full peer review and any attached files.

**Do you want your identity to be public for this peer review?** For information about this choice, including consent withdrawal, please see our Privacy Policy .

Reviewer #1: No

Reviewer #2: No

Reviewer #3: No
---

## [Decision Letter · Decision Letter 1]

5 Jan 2025

Dear Miss 杜,

We are pleased to inform you that your manuscript 'The Japanese encephalitis virus NS1′ protein facilitates virus infection in mosquitoes' has been provisionally accepted for publication in PLOS Neglected Tropical Diseases.

Best regards,

Eng Eong Ooi

Guest Editor

Andrea Marzi

Section Editor

Shaden Kamhawi

co-Editor-in-Chief

Paul Brindley

co-Editor-in-Chief

Please do note the comment from Reviewer #2, which is correct. There is a grammatical error in the short title, which will need to be corrected before formal acceptance or at the proofs stage.

Reviewer's Responses to Questions

**Key Review Criteria Required for Acceptance?**

**Methods**

-Are the objectives of the study clearly articulated with a clear testable hypothesis stated?

-Is the study design appropriate to address the stated objectives?

-Is the population clearly described and appropriate for the hypothesis being tested?

-Is the sample size sufficient to ensure adequate power to address the hypothesis being tested?

-Were correct statistical analysis used to support conclusions?

-Are there concerns about ethical or regulatory requirements being met?

Reviewer #1: (No Response)

Reviewer #2: (No Response)

Reviewer #3: The methods are now clearly described and the RNAseq data is deposited in GEO database

**Results**

-Does the analysis presented match the analysis plan?

-Are the results clearly and completely presented?

-Are the figures (Tables, Images) of sufficient quality for clarity?

Reviewer #1: (No Response)

Reviewer #2: (No Response)

Reviewer #3: Results presented clearly in the revised manuscript

**Conclusions**

-Are the conclusions supported by the data presented?

-Are the limitations of analysis clearly described?

-Do the authors discuss how these data can be helpful to advance our understanding of the topic under study?

-Is public health relevance addressed?

Reviewer #1: (No Response)

Reviewer #2: (No Response)

Reviewer #3: Conclusions valid and clear

**Editorial and Data Presentation Modifications?**

Reviewer #1: (No Response)

Reviewer #2: (No Response)

Reviewer #3: No further comments from me

**Summary and General Comments**

Reviewer #1: This referee is satisfied with the author's comments.

Reviewer #2: The authors have made all the requested changes. Nonetheless, I recommend they change the short title another time as the new one is not grammatically correct. I suggest "JEV NS1' enhances mosquito infection".

Reviewer #3: Authors have sufficiently addressed my concerns.

PLOS authors have the option to publish the peer review history of their article (what does this mean? ). If published, this will include your full peer review and any attached files.

**Do you want your identity to be public for this peer review?** For information about this choice, including consent withdrawal, please see our Privacy Policy .

Reviewer #1: No

Reviewer #2: No

Reviewer #3: No

---

## [Editor Report · Acceptance letter]

Dear Miss Du,

We are delighted to inform you that your manuscript, "The Japanese encephalitis virus NS1′ protein facilitates virus infection in mosquitoes ," has been formally accepted for publication in PLOS Neglected Tropical Diseases.

Best regards,

Shaden Kamhawi

co-Editor-in-Chief

Paul Brindley

co-Editor-in-Chief
